# WHAT LIMITS AGENTIC SYSTEMS EFFICIENCY?

## ABSTRACT

Large Language Models (LLMs), such as OpenAI-o1 and DeepSeek-R1, have demonstrated strong reasoning capabilities. To further enhance LLM capabilities, recent agentic systems, such as *Deep Research*, incorporate web interactions into LLM reasoning to mitigate uncertainties and reduce potential errors. However, existing research predominantly focuses on reasoning performance, often neglecting the efficiency of agentic systems. In this work, we present a comprehensive empirical study that identifies efficiency bottlenecks in web-interactive agentic systems. We decompose end-to-end latency into two primary components: LLM API latency and web environment latency. We conduct a comprehensive empirical study across 15 models and 5 providers to demonstrate high variability in API-based agentic systems. We observe that web environment latency can contribute as much as 53.7% to the overall latency in a web-based agentic system. To improve latency, we propose `SpecCache`, a caching framework augmented with speculative execution that can reduce web environment overhead. Extensive evaluations on two standard benchmarks demonstrate that our approach reduces web environment overhead by up to $3.2\times$, without compromising the performance of the agentic system.

## 1 INTRODUCTION

Large Language Models (LLMs) have become a cornerstone of modern artificial intelligence, achieving outstanding performance across various downstream tasks. Their strengths in natural language understanding Wang et al. (2018), and text generation Clark et al. (2019); Zellers et al. (2019) have enabled significant breakthroughs across disciplines. To further enhance performance, recent advances in large-scale reinforcement learning (RL) have enabled large language models to demonstrate strong long-horizon reasoning abilities. As examples, OpenAI-o1 OpenAI (2025c) and DeepSeek-R1 Guo et al. (2025) leverage RL methods like PPO Schulman et al. (2017) and GRPO Shao et al. (2024) to strengthen problem-solving capabilities, equipping them for complex reasoning tasks Hendrycks et al. (2021); Phan et al. (2025).

Although reasoning models can generate step-by-step reasoning chains, their reasoning processes remain constrained by insufficient knowledge Ji et al. (2023); Rawte et al. (2023). To address this limitation, recent work has proposed agentic systems that combine web interaction with LLM-based reasoning to retrieve external knowledge and access up-to-date information OpenAI (2025b). Existing web-interactive agentic systems can be categorized into the following two types: (1) employing prompt engineering to inject external knowledge into LLMs for complex task completion Li et al. (2025a); Wu et al. (2025b); (2) leveraging reinforcement learning to integrate search capabilities into LLMs Chen et al. (2025); Song et al. (2025); Jin et al. (2025). While existing web-interactive agentic systems primarily focus on improving the reasoning capabilities of LLMs for complex tasks Wei et al. (2025); Wu et al. (2025b), they largely neglect *system efficiency*. The system efficiency (or latency) of agentic systems is critical for applications with low-latency service-level objectives (SLOs), as it directly affects service reliability and user satisfaction Wang et al. (2012); Dean & Barroso (2013).

To address this gap, we systematically benchmark the end-to-end latency of web-interactive agentic systems. As an example, consider two queries from the *WebWalkerQA* Wu et al. (2025b) and *Frames* Krishna et al. (2024) benchmarks to evaluate the latency of a single iteration of the Reflexion-based agentic system Shinn et al. (2023). As shown in Figure 1, both the LLM API and web environment (up to 53.7%) contribute substantially to the latency of web-interactive agentic systems. Accordingly, we separately analyze the latency introduced by the LLM API and the web environment. To identify key contributors to LLM API latency, we analyze the impact of several factors, including

model size (e.g., 70B vs. 405B), API servicing tier (default vs. priority), query date, and output token length across 15 models from 5 providers: Anthropic, DeepSeek, Google, OpenAI, and Together AI (§2.1). To measure web environment latency, we use the *WebWalkerQA* benchmark Wu et al. (2025b), which centers on queries related to international organizations, conferences, and educational institutions, making it well-suited for assessing information retrieval performance (§2.2).

Our empirical analysis reveals the following important observations: (1) High variability across 15 models and 5 providers is observed in LLM response latency. Latency for fixed-length requests may differ by up to $69.21\times$ based on the time they are issued (§2.1); (2) LLM response latency variance persists across dates and locations (§2.1); (3) Web environment latency can contribute as much as $53.7\%$ to the overall latency of agentic systems (§2.2).

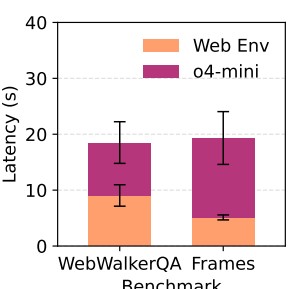

Motivated by these observations, we note that recent advances in API reliability, including OpenAI's priority processing feature[1], have been shown to reduce LLM latency and variance, and we expect such infrastructure-level improvements to continue alleviating this bottleneck over time. In this paper, we focus on the web-environment latency bottleneck of agentic systems and propose a caching strategy to mitigate it, thereby improving the overall latency of agentic systems.

Figure 1: Average latency breakdown per iteration of a Reflexion-based agentic system Shinn et al. (2023) for sampled question answering.

To reduce the web environment latency, we propose `SpecCache` (§3), a novel caching framework that uses speculative execution Leviathan et al. (2023) to mitigate latency in web environments. Specifically, `SpecCache` implements a caching mechanism that stores LLM-generated actions. `SpecCache` introduces a speculative execution path, which uses a draft model to predict the LLM's next action and proactively populate the action cache. Using a draft model unlocks a new dimension that allows environment interaction costs to be concealed by *overlapping* them with model reasoning. Furthermore, `SpecCache` is designed upon the ReAct Yao et al. (2023) abstraction; therefore, `SpecCache` can be applied to not only web-interactive agentic systems but also other turn-based agentic systems that interact with external environments.

In summary, our key contributions are as follows:

- We present a comprehensive end-to-end latency analysis of web-interactive agentic systems, decomposing latency into LLM API and web environment components. Our findings show that both contribute significantly to overall latency.

- To reduce web environment overhead, we propose `SpecCache`, a caching framework that stores a small set of LLM-generated actions and corresponding results. In addition, `SpecCache` employs a model-driven strategy to enable the overlap of environment interaction costs with model reasoning.

- We conduct extensive experiments to demonstrate the effectiveness and efficiency of `SpecCache`. Compared with existing agentic systems on *WebWalkerQA* and *Frames*, `SpecCache` achieves up to a $3.2\times$ reduction in web environment overhead while maintaining performance. Our method does not change the results produced by the agentic system, as the caching framework operates on a separate path and does not interfere with the backbone LLM or the agentic system's reasoning path.

## 2 LATENCY ANALYSIS

Given that both the LLM API and the web environment significantly contribute to the latency of web-interactive agentic systems (Figure 1), we analyze their impacts separately to gain deeper insight. We begin with a detailed examination of LLM API latency in §2.1, followed by an analysis of web environment performance in §2.2.

---

[1] OpenAI Priority Processing: https://openai.com/api-priority-processing/

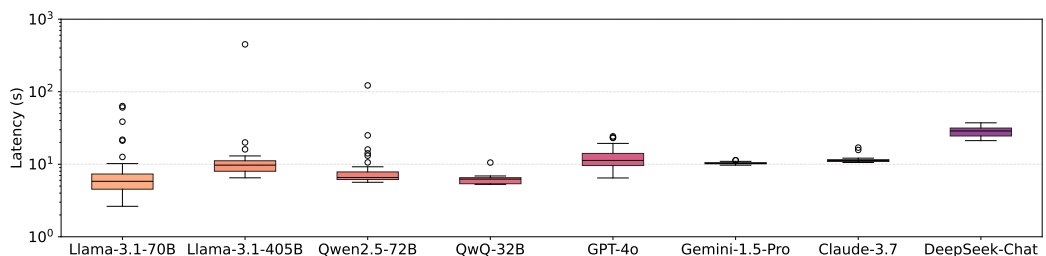

Figure 2: In this figure, we evaluate the end-to-end latency of API calls offered by five AI companies by querying the LLMs every hour. The evaluated models include: (i) Together AI: Llama-3.1-70B, Llama-3.1-405B, Qwen2.5-72B, QwQ-32B; (ii) OpenAI: GPT-4o; (iii) Google: Gemini-1.5-Pro; (iv) Anthropic: Claude-3.7-Sonnet. (v) DeepSeek: DeepSeek-Chat. This figure shows that LLM API response times exhibit high variance, including occasional outliers.

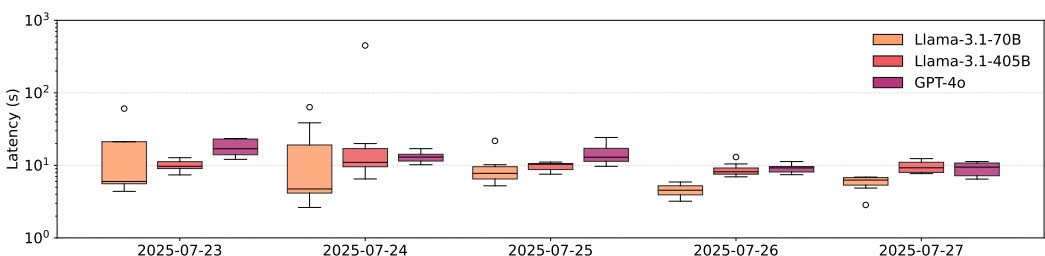

Figure 3: In this figure, we evaluate the end-to-end latency of API calls across different dates. Due to space constraints, we report results from three representative models: Llama-3.1-70B and Llama-3.1-405B provided by Together AI, and GPT-4o from OpenAI. This figure illustrates latency variance over time for various models, with fixed input prompts and a uniform output length.

## 2.1 LLM API

Although many popular models, such as LLaMA Touvron et al. (2023), Qwen Yang et al. (2024b), and DeepSeek Liu et al. (2024), are open-weight, most agentic systems access LLMs through APIs in practice for two primary reasons. First, leading models, such as OpenAI's GPT-4o Hurst et al. (2024), Anthropic's Claude 3.5 Anthropic (2025b), and Google's Gemini 2.5 Google (2025b), remain closed-source and are accessible only via proprietary APIs. Second, the substantial cost and technical complexity of deploying and operating LLMs at scale pose a major barrier for agentic system users Armbrust et al. (2010); Jonas et al. (2017). Therefore, in this section, we monitor API call latency over one week to evaluate its impact on agentic system performance.

**Setup.** Our experiments evaluate LLM API calls from the following providers and their respective models: (i) Anthropic Anthropic (2025a): Claude-3.7-Sonnet, (ii) DeepSeek DeepSeek (2025): DeepSeek-Chat, (iii) Google Google (2025a): Gemini-1.5-Pro, (iv) OpenAI OpenAI (2025a): GPT-4o, and (v) Together AI AI (2025): Llama-3.1-70B, Llama-3.1-405B, Qwen2.5-72B, and QwQ-32B. Unless otherwise specified, all experiments use identical input questions (listed in Appendix A), generate up to 512 output tokens, and are conducted with top-p = 1 and temperature = 0. All experiments are conducted on a CloudLab Duplyakin et al. (2019) instance from Wisconsin. Experiments using priority processing were conducted from September 5 to 7, 2025, as this feature is newly introduced by OpenAI. All other experiments were performed earlier, between July 23 and 27, 2025.

**High Variability in Latency.** We begin with a five-day study evaluating the end-to-end latency of LLM APIs across providers, including Together AI AI (2025), OpenAI OpenAI (2025a), Google Google (2025a), and Anthropic Anthropic (2025a). Each provider is called once per hour for five consecutive days to collect measurement data. Figure 2 illustrates the considerable variance in API latency. For example, the response time for Llama-3.1-405B provided by Together AI AI (2025) ranges from 6.50 seconds to 449.89 seconds. As shown in Figure 3, LLM API call latency

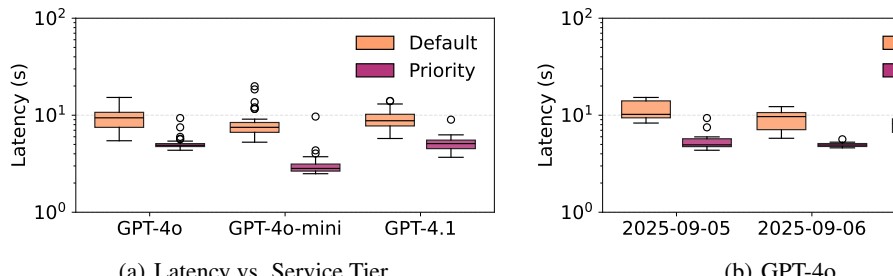

(a) Latency vs. Service Tier  (b) GPT-4o

Figure 4: In this figure, we evaluate the end-to-end latency of API calls under both default and priority tiers across a range of models and dates. The left figure displays the end-to-end latency of various models under both default and priority tiers, while the right figure shows the end-to-end latency of the GPT-4o model across different dates.

exhibits variance over all five days, with fluctuations differing from day to day. High variance in LLM API call latency may arise from constrained GPU resources on the provider side, leading to queuing delays Sheng et al. (2024), or from performance noise in the cloud infrastructure hosting the model De Sensi et al. (2022); Sinha et al. (2022). Due to the variability in LLM API call latency, larger models can occasionally exhibit lower latency than smaller ones. For example, on July 24, 2025, Llama-3.1-405B had lower latency than Llama-3.1-70B. While Gemini-1.5-Pro maintains low variability (3.71%, coefficient of variation), the pronounced variability in Llama-3.1-70B and GPT-4o (135.21% and 36.81%, respectively) poses challenges for consistent performance in latency-sensitive tasks, including language agentic systems Yao et al. (2023); Shinn et al. (2023) and code generation Chen et al. (2021) which rely on LLM APIs.

**Priority Processing.** To reduce the latency and variance of API calls, we conducted a three-day study evaluating the end-to-end latency of OpenAI API calls using priority processing, which is a feature provided by OpenAI. As shown in Figure 4, priority processing effectively reduces both the latency and variance of LLM API calls. Specifically, for GPT-4o, the coefficient of variation in latency decreases from $26.06\%$ (default) to $15.85\%$ (priority), while the average latency drops from 9.39s to 5.08s.

Due to limited space, we present more experimental results from varying model types, request locations, and number of output tokens in Appendix E. The following insights are derived from our extensive experiments: (1) With the fixed input question and output tokens, end-to-end API latency can vary by up to $69.21\times$, resulting in an unstable user experience; (2) Moreover, we observe variability in LLM API latency across different dates and three geographic regions. Specifically, the coefficient of variation in latency for Llama-3.1-70B API calls is $135.21\%$ in Wisconsin, $42.61\%$ in South Carolina, and $106.40\%$ in Utah. (3) To mitigate end-to-end latency and variance, one possible approach is to leverage OpenAI's priority processing feature. However, the priority tier incurs a higher cost than the default tier.

## 2.2 WEB ENVIRONMENT

In this section, we conduct a detailed analysis of the performance characteristics of external tool APIs and web crawlers. Our analysis reveals that these components introduce substantial overhead, potentially reducing deployment efficiency and diminishing user experience. Moreover, our analysis provides the empirical foundation for the caching and prefetching methodology introduced in §3.

**Setup.** To understand the performance trade-offs for web-interactive agentic systems operating in a real-world environment, we ground our analysis in a practical case study. While gym-like environments such as WebArena Zhou et al. (2023) are valuable for reproducibility, they abstract away the noise and performance variability inherent in deploying a live web-interactive language agentic system. Therefore, we utilize the *WebWalkerQA* benchmark Wu et al. (2025b), which requires agentic systems to perform multi-step reasoning and synthesize answers by exploring multiple pages across various real-world websites. The benchmark's focus on knowledge-intensive domains, such as

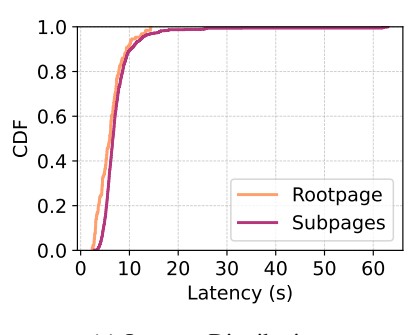 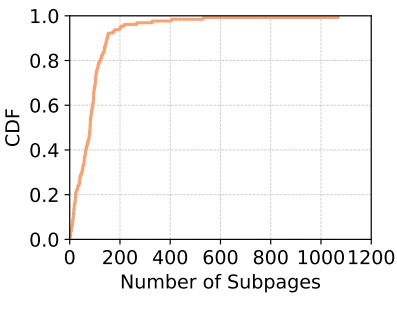

(a) Latency Distribution  (b) Subpages Distribution

Figure 5: The above two CDF figures illustrate the performance characteristics of the web environment from the *WebWalkerQA* benchmark Wu et al. (2025b). (a) The distribution of latency when fetching root URLs and subpages, highlighting the initial overhead. (b) The distribution of the number of clickable subpages available from a root URL, showing a large action space.

international organizations, conferences, and educational institutions, makes it an ideal testbed for evaluating information retrieval performance under realistic conditions. We analyze the performance of a Reflexion-based agentic system Shinn et al. (2023) using QwQ-32B as the backbone reasoning model, following the setup in Wu et al. (2025b). Due to resource constraints, we sample 30 tasks from distinct root domains for this case study.

**Web Crawl Latency Limits System Performance.** An agentic system's interaction cycle in a ReAct Yao et al. (2023) or Reflexion-based agentic system Shinn et al. (2023) is composed of both reasoning (LLM inference) and action (web retrieval). We separately profiled the time spent on reasoning and on actions. Figure 5(a) shows the distribution of latencies for fetching and parsing the HTML of root URLs in our task sample (this includes various conference domains such as sigchi.org, international organization domains such as apec.org, and game producer websites such as rovio.com). As shown in Figure 5(a), the median latency (consisting of both network fetching and HTML parsing) is approximately 6 seconds, with a long tail extending to much higher values, accounting for as much as $45.6\%$ of the total runtime of the agentic system. One potential solution is to use caching techniques Brin & Page (1998) to reduce web crawl latency. However, as shown in Figure 5(b), the large and diverse space of subpages presents significant challenges for effective caching. Given these challenges, we introduce `SpecCache` in the next section as a solution for reducing web environment latency.

## 3 SPECCACHE

In this section, we will first outline the detailed challenges in designing a caching system aimed at reducing web environment overhead for agentic systems (§3.1). Next, we propose a caching framework that reduces the environment interaction cost by enabling parallelism between model inference and environment interaction, while preserving the original trajectory of the agentic system (§3.2). Finally, we provide a detailed discussion of our caching framework (§3.3).

### 3.1 CHALLENGES

In standard LLM agentic system deployments, the agentic system waits for an environment response (e.g., a web page load) before invoking the LLM for the next reasoning step, and vice versa. To improve efficiency, our goal is to hide this environment interaction cost by *overlapping it* with model reasoning. A natural approach is to develop a caching mechanism that prefetches environment responses likely to be needed in future steps. However, designing an effective cache for language agentic systems is non-trivial due to the sheer size of the action space. For example, our analysis of the *WebWalkerQA* dataset Wu et al. (2025b) reveals that each of the 138 root pages contains a median of 81 clickable subpages (Figure 5(b)), representing possible next actions. This high branching factor makes it difficult to accurately anticipate which observations will be needed, presenting a

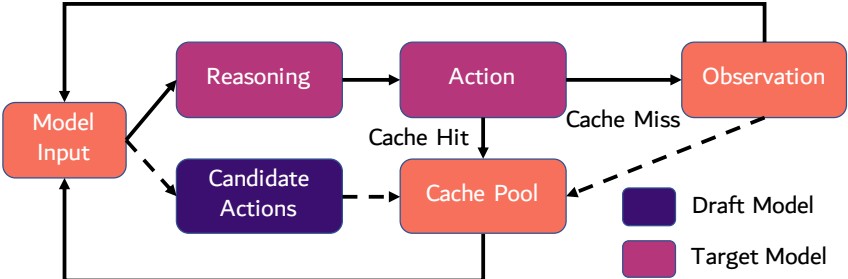

Figure 6: This figure shows the workflow of our `SpecCache` framework. In each iteration, the model input is fed to two independent and non-blocking threads, one Reflexion-based thread and one caching thread aimed at generating candidate actions. The caching thread updates the cache pool with its candidate actions. When the Reflexion-based thread selects an action, it first queries the cache pool. If the cache misses, it executes the action, retrieves the corresponding observation, and proceeds to the next iteration, updating the cache pool with the new action-observation pair.

key challenge for prefetching and caching strategies aimed at improving runtime efficiency. Naive strategies, such as uniform action sampling, would result in a near-zero cache hit rate.

## 3.2 CACHING FRAMEWORK

In this section, we propose a caching and prefetching framework that decouples and overlaps model reasoning and environment interaction, significantly reducing wall-clock latency without compromising task success rates. Our method is a caching system that employs a model-based action-observation cache.

**Action-Observation Cache.** The active-observation cache, following an LRU policy, is designed to store the outcomes of specific actions taken from a given state (e.g., a webpage). When the target LLM decides on an action, it first queries this cache. A cache hit signifies that the action has been previously executed, and the corresponding observation is immediately retrieved, bypassing the costly interaction with the environment. This on-demand caching of action-observation pairs is crucial for accelerating interactions within a session.

**Model-Based Prefetching.** To build the action-observation cache, we introduce a model-based prefetching scheme. This component of our framework moves beyond reactive caching to proactively explore and cache potential future states. Leveraging ideas from speculative execution Chen et al. (2023); Leviathan et al. (2023), we use a draft model, a smaller LLM running asynchronously with the primary reasoning LLM (the target model). The role of the draft model is to predict the future actions that the target model is likely to take from the current state.

The prefetching process unfolds as follows:

1. **Asynchronous Action Prediction:** While the target LLM performs reasoning, the draft model generates candidate actions (e.g., web crawls), which are executed in parallel.

2. **Asynchronous Caching:** The observations resulting from these speculative actions are stored in the action-observation cache.

When the target LLM eventually determines its next action, it first consults the cache. If the draft model's prediction is accurate, the observation is already present, and the agentic system can proceed instantaneously. This asynchronous prefetching effectively decouples the agentic system's reasoning from the environment's response time (dashed lines in Figure 6), enabling the design of more efficient agentic systems.

## 3.3 DISCUSSION

Our speculative caching approach introduces new trade-offs that balance latency reduction with increased compute and environment load. The draft model introduces additional computation for

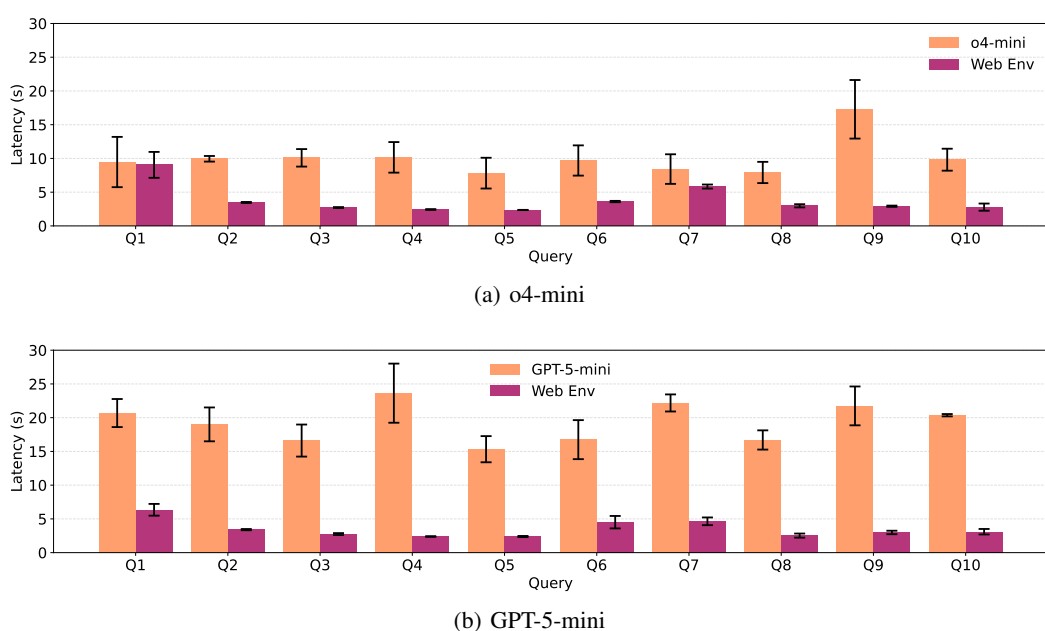

(a) o4-mini

(b) GPT-5-mini

Figure 7: This figure shows the iteration-wise latency breakdown of the Reflexion-based agentic system using o4-mini and GPT-5-mini as backbone models, evaluated on sampled questions from *WebWalkerQA* Wu et al. (2025b). We perform five runs for each sampled question. The sampled questions are listed in Appendix C.

running speculative rollouts asynchronously. We complete speculative actions even after the target model selects its next move. This preserves useful data in the cache for future steps. In cases where speculative actions are not used, the main agentic system flow is not interfered with.

The principles underpinning our caching and prefetching framework are not limited to web-interactive agentic systems. This methodology can be generalized to any turn-based agentic system that operates in an environment where the feedback loop constitutes a significant portion of the overall latency. By decoupling reasoning from interaction and proactively exploring the action space, our approach provides a robust and scalable solution for accelerating a wide range of language-agentic system applications.

## 4 EXPERIMENTS

In this section, we begin by detailing our experimental setup (§4.1). We then evaluate our framework's performance on web-based reasoning and retrieval tasks, analyzing its effectiveness in reducing end-to-end latency in real-world agentic system deployments (§4.2).

### 4.1 SETUP

**Models.** We use o4-mini and GPT-5-mini as target models in a Reflexion-based agentic system to evaluate and control for the effects of model variation. These models have demonstrated state-of-the-art performance on web exploration benchmarks, outperforming open-sourced models such as Qwen2.5-72B Sun et al. (2025); Li et al. (2025a;b); Wu et al. (2025a). For speculative execution, we employ GPT-4.1-mini as draft models. As all models are provided by OpenAI, we use priority processing to reduce the latency and variance of LLM API calls.

**Benchmarks.** Reasoning agentic systems capable of multi-hop, in-depth exploration of real-world web content remain a challenging research area, despite recent progress enabled by more powerful models Hurst et al. (2024); Anthropic (2025a); Yang et al. (2024a). As highlighted in Wu et al. (2025b), existing benchmarks such as GAIA Mialon et al. (2023), MMinA Zhang et al. (2024), and

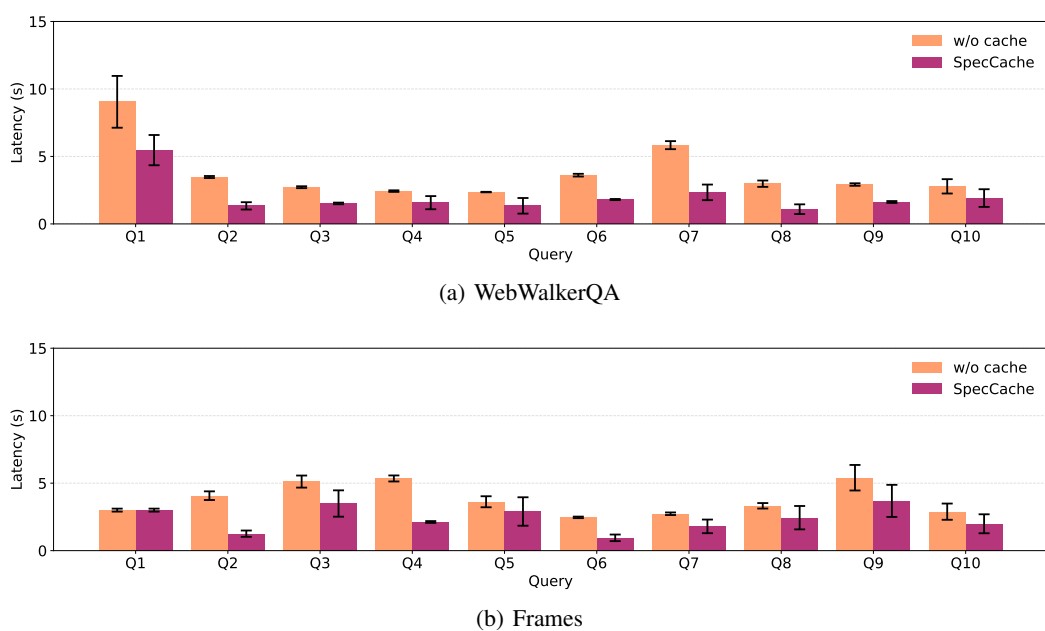

(a) WebWalkerQA

(b) Frames

Figure 8: This figure shows the iteration-wise latency for the agentic systems accelerated by `SpecCache` when answering sampled questions from *WebWalkerQA* Wu et al. (2025b) and *Frames* Krishna et al. (2024). We use o4-mini as the target model and GPT-4.1-mini as the draft model. We perform five runs for each sampled question. The sampled questions are listed in Appendix C.

AssistantBench Yoran et al. (2024) primarily focus on breadth-wise reasoning, and do not sufficiently evaluate depth-wise web exploration capabilities.

We conduct experiments on two benchmarks designed to capture both multi-hop reasoning and in-depth web exploration: *WebWalkerQA* Wu et al. (2025b) and *Frames* Krishna et al. (2024).

- *WebWalkerQA* evaluates an agentic system's ability to perform multi-hop web reasoning over a large set of websites. We sample a query from each distinct root URL for our evaluations, where the agentic system is provided with the root URL as a starting point.
- *Frames* is a benchmark consisting of factual questions that require synthesizing information from 2 to 15 Wikipedia pages. To emphasize the multi-hop setting, we select a subset of queries that require information from at least 5 distinct sources. The agentic system is provided with only a single Wikipedia page as the seed URL.

We cap the maximum number of iterations per task at 10, where each iteration consists of a reasoning step, an action step, and a critique step. Empirically, most tasks are completed within 5-6 iterations. Given budget constraints, we sample 10 questions from each benchmark to analyze the LLM and web environment overhead in agentic systems, as well as the acceleration achieved through `SpecCache`.

**Metric.** We measure the agent latency for each Reflexion-based iteration across multiple workloads, averaging results over five runs. Each iteration latency includes the time for observation extraction, tool use, and reflection within a complete Reflexion agent cycle. Averaging mitigates noise and inter-run variance from model APIs, allowing us to better isolate environmental bottlenecks and quantify the overhead reductions achieved by our approach.

### 4.2 EXPERIMENTAL RESULTS

Figure 7(a) presents the iteration-wise latency breakdown of o4-mini and the web environment for each question sampled from *WebWalkerQA* Wu et al. (2025b). From Figure 7(a), we observe high API latency variance for o4-mini, and the web environment constitutes a major component of the

Reflexion-based agentic system, consistent with our empirical findings. We also use GPT-5-mini as the target model, and the results are shown in Figure 7(b). GPT-5-mini is more powerful than o4-mini but introduces greater overhead to agentic systems. However, web environment latency can still account for up to 23.5% of the overall end-to-end latency. While priority processing helps mitigate latency and variance in LLM API calls, the variance in Figure 7 is primarily due to the variation in input and output tokens for each run.

We also present the acceleration achieved by the proposed `SpecCache` in Figure 8. As shown in Figure 8, `SpecCache` achieves up to a $3.2\times$ reduction in web environment latency for answering sampled questions. Importantly, our solution always improves the overall efficiency of agentic systems and never adds overhead to their end-to-end latency. These results reveal a new axis for accelerating agentic systems: allocating more compute to asynchronous assistant models allows environment overhead to be overlapped with LLM reasoning.

Due to limited space, additional experimental results are provided in Appendix F, which are consistent with the observations above.

## 5 RELATED WORK

**Large Language Models.** The Transformer architecture Vaswani et al. (2017) has been successfully applied to a wide variety of tasks, including text classification Wang et al. (2018); Sarlin et al. (2020), text generation Zellers et al. (2019); Sakaguchi et al. (2021), mathematical reasoning Cobbe et al. (2021); Hendrycks et al. (2021), and code generation Austin et al. (2021). The development of GPT models Brown et al. (2020) highlights how scaling up language models substantially improves their performance across a range of downstream tasks. Inspired by the success of GPT, several large-scale language models have been introduced, including LLaMA Touvron et al. (2023), Gemma Team et al. (2024), Qwen Yang et al. (2024b), and DeepSeek Liu et al. (2024); Guo et al. (2025).

**Web-Interactive Agentic Systems and Benchmarks.** Recent web-interactive agentic systems, including Search-o1 Li et al. (2025a), ReSearch Chen et al. (2025), Search-R1 Jin et al. (2025), and WebDancer Wu et al. (2025a), enhance the reasoning capabilities of large language models by integrating web interaction into their decision-making. Concurrently, benchmarks such as GAIA Mialon et al. (2023), MMinA Zhang et al. (2024), AssistantBench Yoran et al. (2024), BrowseComp Wei et al. (2025), and WebWalker Wu et al. (2025b) have been proposed to evaluate the performance of agentic systems in real-world web environments.

## 6 LIMITATIONS AND FUTURE WORK

First, we primarily focus on reducing latency arising from the web environment. Although we leverage priority processing provided by OpenAI to reduce LLM API latency and its variance, it remains unclear whether such latency and variance can be effectively reduced from the user side. Secondly, both the number of rounds and the total tokens generated per round pose major bottlenecks to agentic system efficiency. Future work will explore strategies to reduce these overheads. Lastly, we believe a deeper dive into the API traffic analysis would be of independent research interest, shedding light on how much request batching, query priority scheduling, and LLM execution contribute to the end-to-end latency and variance, respectively.

## 7 CONCLUSION

In this paper, we provide a comprehensive empirical analysis of web-interactive agentic systems. Our findings reveal that both the LLM API and the web environment significantly contribute to agentic system latency. To reduce agentic system latency, we propose `SpecCache`, a caching technique designed to mitigate web environment overhead. Extensive evaluations demonstrate that `SpecCache` reduces web environment latency by up to $3.2\times$.

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

## A    LATENCY TEST QUESTION.

This section introduces the question used to measure LLM API latency, with a fixed input and a controlled number of output tokens.

---

**QUESTION:**

Tell a story about Blackberry. Make the story detailed, with rich descriptions, character development, and dialogue. Aim for a story that would take at least $n$ tokens to tell.

---

Table 1: The question used to measure LLM API latency.

where $n$ can be set to 64, 128, 256, 512, or 1024, depending on the number of output tokens.

## B    EXAMPLES OF MATH QUESTIONS

This section presents the sampled math questions used to measure LLM API latency.

---

**QUESTION 1:**

Find the constant term in the expansion of

$$\left(10x^3 - \frac{1}{2x^2}\right)^5$$

---

**QUESTION 2:**

At what value of $y$ is there a horizontal asymptote for the graph of the equation
$y = \frac{4x^3+2x-4}{3x^3-2x^2+5x-1}$?

---

**QUESTION 3:**

How many zeroes are at the end of $42!$ (42 factorial)? (Reminder: The number $n!$ is the product of the integers from 1 to $n$. For example, $5! = 5 \cdot 4 \cdot 3 \cdot 2 \cdot 1 = 120$.)

---

**QUESTION 4:**

Suppose that $ABCD$ is a trapezoid in which $\overline{AD}||\overline{BC}$. Given $\overline{AC} \perp \overline{CD}$, $\overline{AC}$ bisects angle $\angle BAD$, and $[ABCD] = 42$, then compute $[\triangle ACD]$.

---

Table 2: The sampled math questions used to measure LLM API latency.

# C    SAMPLED QUESTIONS

## C.1    WEBWALKERQA

In this section, we present the sampled WebWalkerQA Questions used to measure LLM and Web API latency in Table 3.

---

**QUESTION 1:**

During the 7th China International Import Expo (CIIE) in 2024, when will the National Exhibition and Convention Center (Shanghai) be closed for security measures, and who is permitted to access the venue during this period?

**QUESTION 2:**

Who were the recipients of the POMS Fellows Award in 2006 and the Wickham Skinner Award for Teaching Innovation in 2018?

**QUESTION 3:**

For ACL 2024, what is the deadline for students requiring financial assistance to apply for discounted virtual registration, and by what date will they be notified about the selection for D&I subsidies?

**QUESTION 4:**

What was the specific schedule for the social event held on the evening after the ACL 2023 best paper awards ceremony?

**QUESTION 5:**

When is the paper submission deadline for the ACL 2025 Industry Track, and what is the venue address for the conference?

**QUESTION 6:**

Where is the Web Conference 2024 Welcome Reception held and what is the nearest transportation method from the Resorts World Convention Centre?

**QUESTION 7:**

Which event has a higher total reward pool, the SHIBUYA Y3K event on October 2, 2024, or the upcoming The Smurfs: Gargamel's Castle experience?

**QUESTION 8:**

What is the official launch date of Junkworld on Apple Arcade, and what new feature was introduced in the January 2024 update?

**QUESTION 9:**

Find the first IGG recruitment contact email in Asia in alphabetical order.

**QUESTION 10:**

Who was the chair of the 12th APEC Tourism Ministerial Meeting held in Urubamba on June 9, 2024?

---

Table 3: The sampled WebWalkerQA Questions used to measure LLM and Web API latency.

## C.2 FRAMES

In this section, we present the sampled Frames Questions used to measure LLM and Web API latency in Table 4.

---

**QUESTION 1:**

I have an element in mind and would like you to identify the person it was named after. Here's a clue: The element's atomic number is 9 higher than that of an element discovered by the scientist who discovered Zirconium in the same year.

---

**QUESTION 2:**

As of July 1, 2024, what is the parent company of the current record label of the singer of Edge of Seventeen?

---

**QUESTION 3:**

According to the 1990 United States census, what was the total population of the cities in Oklahoma that had at least 100,000 residents according to the 2020 United States census?

---

**QUESTION 4:**

The oldest extant football team in Italy plays in a stadium. The stadium is named after a person. Who was the emperor of China when that person was 5 years old?

---

**QUESTION 5:**

Of the four main characters on Seinfeld, which actor is the oldest?

---

**QUESTION 6:**

Which species from the genus mulona are both found in the same country?

---

**QUESTION 7:**

I am moving to the G40 postcode area - what train stations are nearby, as of 2024?

---

**QUESTION 8:**

Which player scored more than 15 goals in Eredevisie during the 21-22 season and had previously played for Auxerre?

---

**QUESTION 9:**

Which MP standing as the leader of a major party in the 2019 United Kingdom General Election was also an MP for Henley?

---

**QUESTION 10:**

What is the etymology of the name of the province to the east of the province in which Hazrati Sultan District is located?

---

Table 4: The sampled Frames Questions used to measure LLM and Web API latency.

# D PROMPTS AND TRAJECTORY

## D.1 TARGET LLM PROMPTS

We evaluate `SpecCache` on top of a Reflexion Shinn et al. (2023) agentic system. Table 5-7 outline the prompts used for each component of Reflexion.

---

**TARGET MODEL ACTION PROMPT:**

Digging through the buttons to find quality sources and the right information. You have access to the following tools:
<Tool Description>
Use the following format:
Question: the input question you must answer
Thought: you should always think about what to do
Action: the action to take, should be one of [<Tool Names>]
Action Input: the input to the action
Observation: the result of the action
Action: the action to take, should be one of [<Tool Names>]
Action Input: the input to the action
Observation: the result of the action
... (this Thought/Action/Action Input/Observation can be repeated zero or more 20 times)
Notice:
- You must take action at every step. When you take action, you must use the tool with the correct format and output the action input.
- You can not say "I'm sorry, but I cannot assist with this request."!!! You must explore.
- When you have sufficient information to answer the query, provide your final answer in the format: "Final Answer: <your answer>"
- Action Input should be valid JSON.
- IF YOU DO NOT HAVE SUFFICIENT INFORMATION, CONTINUE EXPLORING BY TAKING ACTION.
- YOU MUST TAKE ACTION AT EVERY STEP UNLESS YOU ARE PRODUCING YOUR FINAL ANSWER. WHEN YOU TAKE ACTION, YOU MUST USE THE TOOL WITH THE CORRECT FORMAT AND OUTPUT THE ACTION INPUT. THEREFORE, YOU MUST OUTPUT AN ACTION AND AN ACTION INPUT.
- IF YOU ARE PRODUCING YOUR FINAL ANSWER, YOU MUST OUTPUT THE FINAL ANSWER IN THE FORMAT: "Final Answer: <your answer>"
Begin!
<Query>

---

Table 5: The prompt for the backbone LLM to take an action.

---

**TARGET MODEL SELF-REFLECTION PROMPT:**

You are an information extraction agent. Your task is to analyze the given observation and extract ANY information that could help answer the query, including:
- Direct facts and data
- Reasoning and conclusions made by the model
- Historical information that could be relevant
- Any insights that contribute to solving the query
- Background knowledge that supports the answer
**Input:**
- Query: "<Query>"
- Observation: "<Current Observation>"
**Output (JSON):**

---

```
{
    "usefulness": true,
    "information": "<Extracted Useful Information> using string format"
}
```
Or, if the observation contains NO potentially useful information at all:
```
{
    "usefulness": false
}
```
**Guidelines:**
- Be generous in what you consider "useful information"
- Include reasoning, conclusions, and background knowledge
- If the observation contains ANY information that could contribute to solving the query, mark it as useful
- Only mark as false if the observation is completely irrelevant
Only respond with valid JSON.

Table 6: The prompt for the target model to perform self-reflection.

**MAIN MODEL EVALUATOR PROMPT:**

You are a query answering agent. Your task is to evaluate whether the accumulated useful information is sufficient to answer the current query with HIGH CONFIDENCE. If it is sufficient and you are very confident in the answer, return a JSON object with a "judge" value of true and an "answer" field with the answer. If the information is insufficient or you have doubts, return a JSON object with a "judge" value of false.
**Input:**
- Query: "<Query>"
- Accumulated Information: "<Accumulated Useful Information>"
**Output (JSON):**
```
{
    "judge": true,
    "answer": "<Generated Answer> using string format"
}
```
Or, if the information is insufficient or you have doubts:
```
{
    "judge": false
}
```
**Guidelines:**
- Only mark as sufficient if you are VERY CONFIDENT in the answer
- If you have any doubts about facts, reasoning, or completeness, mark as insufficient
- Consider whether you need more information to verify your answer
- The answer should be clear, complete, and directly address the query
- When in doubt, prefer to continue exploring rather than give a potentially wrong answer
Only respond with valid JSON.

Table 7: The evaluator prompt for the target model to create internal feedback.

## D.2 DRAFT MODEL PROMPT

In this section, we present the prompt for the draft model to predict action in Table 8.

---

**DRAFT MODEL ACTION PREDICTION PROMPT:**

Digging through the buttons to find quality sources and the right information. You have access to the following tools:

<Tool Description>

Use the following format:

Question: the input question you must answer

Thought: you should always think about what to do

Action: the action to take, should be one of [<Tool Names>]

Action Input 1: {{"button": "About"}}

Action Input 2: {{"button": "Contact"}}

Action Input 3: {{"button": "Application"}}

Observation: the result of the action

Action: the action to take, should be one of [<Tool Names>]

Action Input 1: {{"button": "News"}}

Action Input 2: {{"button": "Info"}}

Action Input 3: {{"button": "Faculty"}}

Observation: the result of the action

... (this Thought/Action/Action Input/Observation can be repeated zero or more 20 times)

Notice:

- You must take action at every step. When you take action, you must use the tool with the correct format and output 3 action inputs.

- You must always output three Action Input lines (Action Input 1, Action Input 2, Action Input 3) for each Action, unless there are fewer than three distinct valid inputs available.

- If there are fewer than three, output as many as are available.

- When you can not find the information you need, you should visit page of previous page recursively until you find the information you need.

- You can not say "I'm sorry, but I cannot assist with this request."!!! You must explore.

- If you do not have sufficient information, continue exploring.

- Action Input should be a valid JSON.

- Do not recommend navigation buttons such as "About Wikipedia", "Search", "Create account", "Log in", "View source", "Print/export", "Navigation".

- Focus on content-specific buttons that are likely to contain information relevant to your query, such as:

   - Names of people, places, events, or topics

   - Years, dates, or time periods

   - Specific categories or sections

   - Links to related articles or detailed pages

Begin!

<Query>

---

Table 8: The prompt for the draft model that performs speculative action prediction in `SpecCache`.

### D.3 Trajectory Example

In this section, we show an example of a trajectory. We use {Action} to indicate a cache hit and {Action} to indicate cache exists and the content will not be retrieved.

**Query:**

When is the paper submission deadline for the ACL 2025 Industry Track, and what is the venue address for the conference?

**Root URL:**

`https://2025.aclweb.org/`

**Draft LLM:**

Thought: The "Call for Industry Track" button will likely provide details on submission deadlines for the Industry Track. For the venue address, "Participants Info" or similar buttons may give explicit venue details. Starting by exploring the "Call for Industry Track" for the specific deadline and the other buttons for venue information.

Action: visit_page

Action Input 1: {"button": "Call for Industry Track"}

Action Input 2: {"button": "Participants Info"}

Action Input 3: {"button": "Program"}

**Target LLM:**

Thought: Question: When is the paper submission deadline for the ACL 2025 Industry Track, and what is the venue address for the conference? Thought: To find the Industry Track deadline, click the {{Call for Industry Track}} button; to find the venue address, click {{Participants Info. }},

Action: visit_page,

Action Input: {"button": "Call for Industry Track"}

Observation: <Content retrieved from `https://2025.aclweb.org/calls/industry_track/`>

**Draft LLM:**

Thought: I have found the paper submission deadline for the Industry Track (21 March 2025). Next, I need to find the venue address. It is most likely in sections such as "Participants Info", "FAQ", or "Visa". I will explore these buttons to locate the venue address.

Action: visit_page

Action Input 1: {"button": "Participants Info"}

Action Input 2: {"button": "FAQ"}

Action Input 3: {"button": "Visa"}

**Target LLM:**

Thought: Question: When is the paper submission deadline for the ACL 2025 Industry Track, and what is the venue address for the conference? Thought: We already extracted the paper submission deadline (21 March 2025) from the Industry Track page. Next, we need to find the venue address by navigating to the {{ Participants Info }} section via the conference menu. Action: visit_page Action Input: {button: Toggle menu} Observation: The conference menu has appeared, showing entries including {{Participants Info.}},

Action: visit_page,

Action Input: {"button": "Participants Info"}

Observation: <Content retrieved from `https://2025.aclweb.org/venue/`>

**Target LLM:**

Final Answer: The paper submission deadline for the ACL 2025 Industry Track is 21 March 2025, and the conference will be held at Austria Center Vienna, Bruno-Kreisky-Platz 1, 1220 Wien, Austria.

# E  ADDITIONAL LLM API LATENCY MEASUREMENT RESULTS

In this section, we present further experimental results from our inference latency measurements. We follow the experimental setup mentioned in Section 2.1.

## E.1  REGIONAL ABLATION STUDY

For the regional ablation study, we use CloudLab Duplyakin et al. (2019) instances located in Madison, Clemson, and Utah. The results are shown in Figure 9. Figure 9 shows that LLM API latency exhibits variance regardless of the city from which requests are issued.

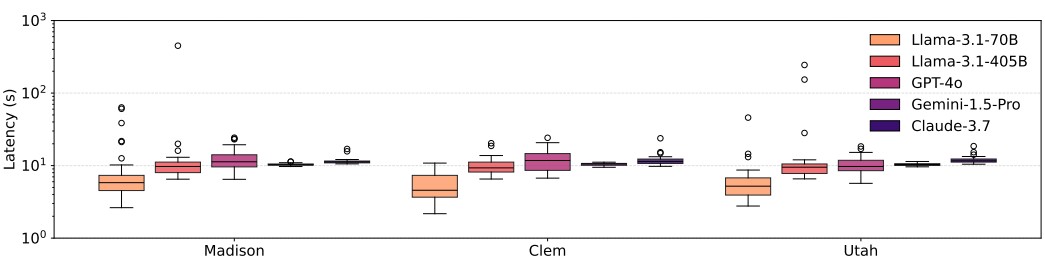

Figure 9: This figure shows the latency variance across regions for models including Llama-3.1-70B, Llama-3.1-405B, GPT-4o, Gemini-1.5-Pro, and Claude-3.7-Sonnet. All requests use the same input and a fixed number of output tokens. Latency is measured by sending requests from machines located in Wisconsin (Madison), South Carolina (Clemson), and Utah.

## E.2  VARY OUTPUT TOKENS

In this section, we measure the end-to-end latency of LLM API calls by fixing the input questions and varying the number of output tokens. The results are shown in Figure 10. As shown in Figure 10, LLM API latency tends to increase with the number of output tokens, although some variance is observed.

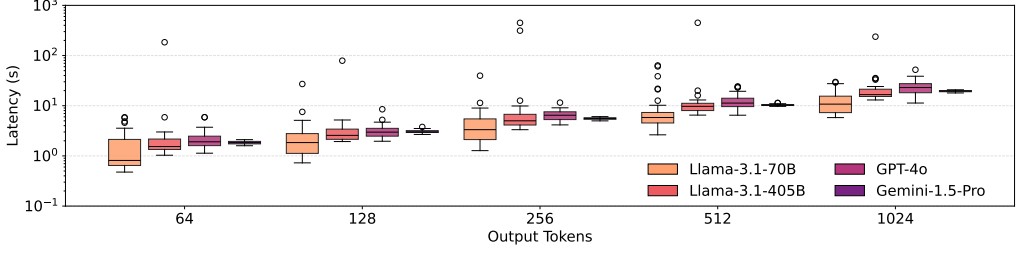

Figure 10: From July 23 to July 27, 2025, we evaluated the end-to-end latency of API calls provided by two AI companies by fixing the input sequence and varying the number of output tokens from 64 to 1024. The evaluated models include: (i) Together AI: Llama-3.1-70B, Llama-3.1-405B; (ii) OpenAI: GPT-4o; (iii) Google: Gemini-1.5-Pro. The figure illustrates an upward trend in LLM API latency with increasing output token count.

### E.3 EXPLORE DIFFERENT MODES

In this section, we evaluate the latency across different execution modes offered by Together AI. As shown in Figure 11, dedicated mode[2] reduces the variance of API calls compared to serverless mode; however, it is also more expensive due to per-minute billing.

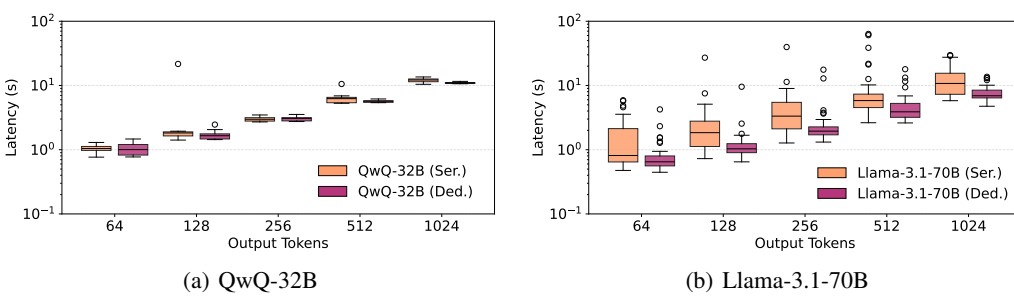

(a) QwQ-32B                          (b) Llama-3.1-70B

Figure 11: This figure illustrates the relationship between API call latency and the number of output tokens under various API deployment modes. QwQ-32B (Ser.) and Llama-3.1-70B (Ser.) denote API calls made in serverless mode, while QwQ-32B (Ded.) and Llama-3.1-70B (Ded.) refer to calls made in dedicated mode.

### E.4 EVALUATE MORE MODELS

In this section, we present additional end-to-end latency measurements of LLM API calls across a broader range of models, as shown in Figure 12. The results in Figure 12 support the conclusion that LLM API calls exhibit high variability.

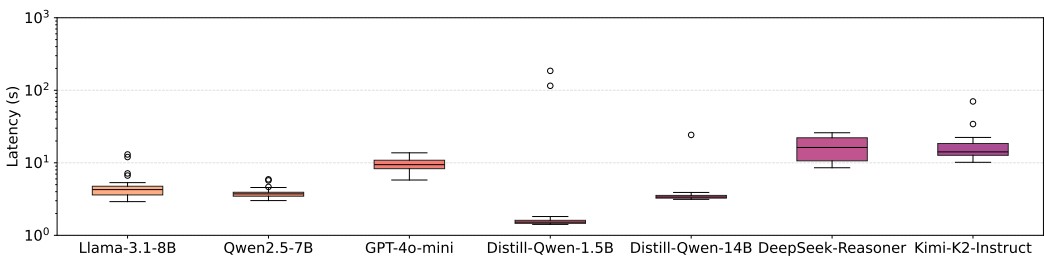

Figure 12: In this figure, we evaluate the end-to-end latency of API calls offered by three AI companies by querying the LLMs every hour. The evaluated models include: (i) Together AI: Llama-3.1-8B, Qwen2.5-7B, DeepSeek-R1-Distill-Qwen-1.5B, DeepSeek-R1-Distill-Qwen-14B, and Kimi-K2-Instruct; (ii) OpenAI: GPT-4o-mini; (iii) DeepSeek: DeepSeek-Reasoner. This figure shows that LLM API response times exhibit high variance, including occasional outliers.

### E.5 OUTPUT TOKENS AFFECT LATENCY

In this section, we sample four questions at random from the MATH dataset Hendrycks et al. (2021) (listed in Appendix B) and permit the LLMs to generate responses of arbitrary length, in contrast to prior experiments that employed fixed token limits. As shown in Figure 13, although QwQ-32B is faster than Qwen2.5-72B per output token (Figure 2), it produces more output tokens due to the reasoning process, leading to higher overall latency. Notably, both QwQ-32B and Gemini-1.5-Pro are reasoning-oriented models. While Gemini-1.5-Pro is slower than QwQ-32B with fixed input and output tokens, it demonstrates greater efficiency on sampled math questions by generating fewer

---

[2]With Together AI, we can create on-demand dedicated endpoints with the following advantages: (1) Consistent, predictable performance, unaffected by other users' load in our serverless environment; (2) No rate limits, with a high maximum load capacity; (3) More cost-effective under high utilization; (4) Access to a broader selection of models.

output tokens per answer. Therefore, learning to generate correct answers using fewer tokens is an important consideration for model training.

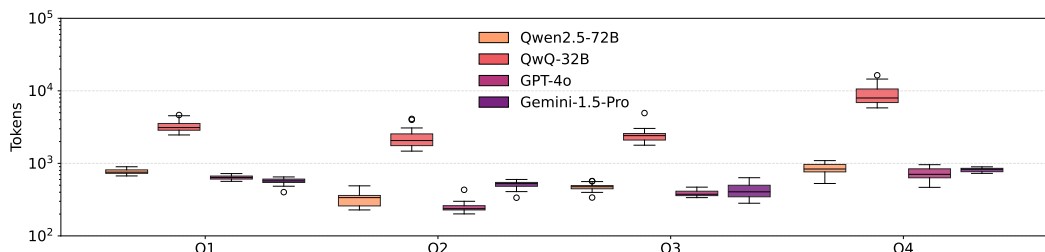

(a) This figure shows the number of tokens generated for answering Q1–Q4 in Appendix B across four models: Qwen2.5-72B, QwQ-32B, GPT-4o, and Gemini-1.5-Pro. It highlights that QwQ-32B produces significantly more output tokens than the others when solving the sampled math problems.

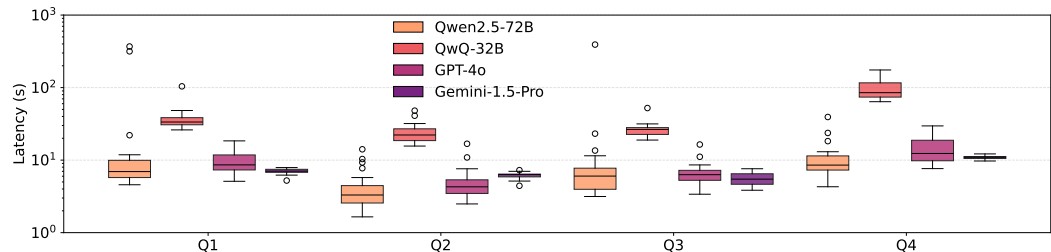

(b) This figure reports the latency for answering Q1–Q4 in Appendix B across four models: Qwen2.5-72B, QwQ-32B, GPT-4o, and Gemini-1.5-Pro. This indicates that QwQ-32B exhibits the highest end-to-end latency among the evaluated models.

Figure 13: This figure shows the LLM API latency and the number of generated tokens for answering Q1–Q4 from Appendix B. Following prior work Brown et al. (2024), we set the temperature to 0.6 and top-p to 0.95 when solving the math problems. The results show that output token length significantly affects LLM API end-to-end latency.

### E.6   ADDITIONAL RESULTS ON PRIORITY PROCESSING

In this section, we present additional results on priority processing, a feature offered by OpenAI. As illustrated in Figure 14, we observe that priority processing consistently reduces end-to-end API call latency across different models, including GPT-4.1 and GPT-4o-mini.

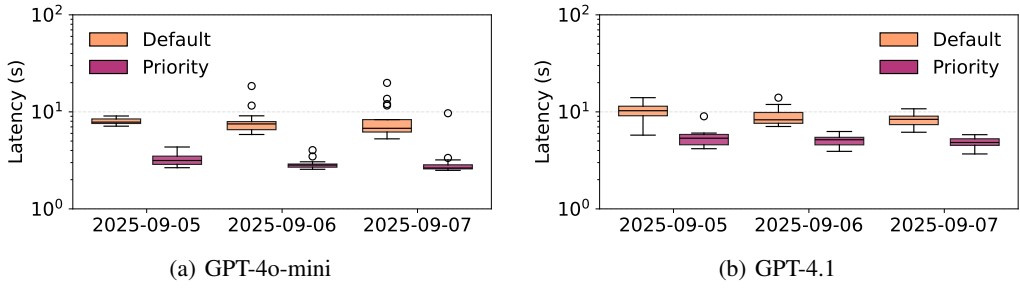

(a) GPT-4o-mini                    (b) GPT-4.1

Figure 14: In this figure, we present an evaluation of end-to-end API call latency for additional models under both default and priority tiers, measured across different dates.

## F    MORE EXPERIMENTAL RESULTS

In this section, we present additional experimental results in Figure 15. Using GPT-5-mini as the target model and GPT-4.1-mini as the draft model, the results demonstrate that `SpecCache` consistently reduces web environment latency across different target models.

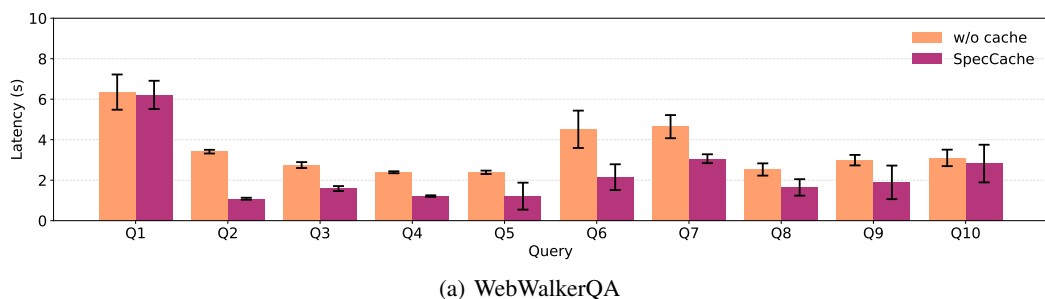

(a) WebWalkerQA

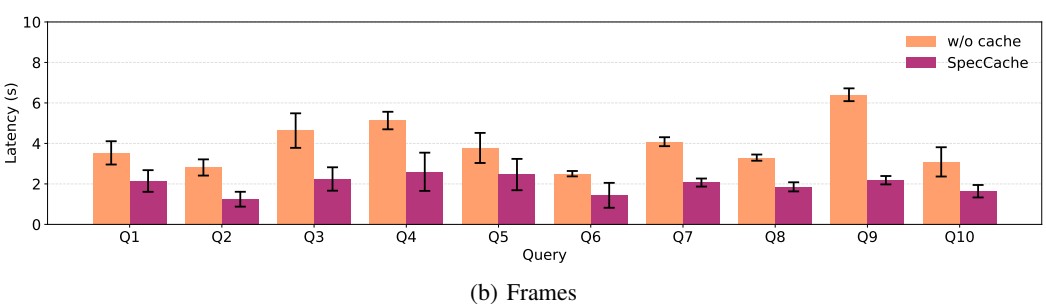

(b) Frames

Figure 15: This figure shows the iteration-wise latency breakdown for the agentic systems accelerated by `SpecCache` when answering sampled questions from *WebWalkerQA* Wu et al. (2025b) and *Frames* Krishna et al. (2024). We use GPT-5-mini as the target model and GPT-4.1-mini as the draft model. We perform five runs for each sampled question. The sampled questions are listed in Appendix C.

## G    MODEL VERSION

Table 9 lists the versions of the models used in this paper.

| Model Name | Version |
| --- | --- |
| GPT-4o-mini | gpt-4o-mini-2024-07-18 |
| GPT-4o | gpt-4o-2024-08-06 |
| GPT-4.1-mini | gpt-4.1-mini-2025-04-14 |
| GPT-4.1 | gpt-4.1-2025-04-14 |
| o4-mini | o4-mini-2025-04-16 |
| GPT-5-mini | gpt-5-mini-2025-08-07 |
| Claude-3.7 | Claude-3-7-Sonnet-20250219 |
| DeepSeek-V3 | DeepSeek-V3-0324 |
| DeepSeek-R1 | DeepSeek-R1-0528 |

Table 9: This table presents the model versions used in this paper.

## H    LLM USAGE

We utilized an LLM to polish the writing by correcting grammar in our completed draft. Importantly, the LLM was not used to survey related work or to propose research ideas.

