# OpenReview forum: "What Limits Agentic Systems Efficiency?"
_ICLR.cc/2026/Conference — Submitted to ICLR 2026_

### Official Review · Reviewer_bymT · 2025-10-18

**Soundness:** 2
**Presentation:** 3
**Contribution:** 2
**Rating:** 2
**Confidence:** 5

**Summary:**

Large Language Models (LLMs) have advanced in reasoning capabilities but face limitations in knowledge, prompting the development of agentic systems that integrate web interactions to access external information. The motivation stems from the neglect of efficiency in these systems, where latency impacts service reliability and user satisfaction in applications with strict SLOs. Challenges include high variability in LLM API latency across models, providers, dates, and locations, as well as significant web environment latency contributing up to 53.7% of total delay due to large action spaces. The solution involves a comprehensive latency analysis and the proposal of SpecCache, a caching framework using speculative execution with a draft model to prefetch actions, overlapping environment interactions with reasoning to reduce web overhead by up to 3.2x without affecting performance.

**Strengths:**

1. Conducts a thorough empirical analysis of end-to-end latency in web-interactive agentic systems. Decomposes latency into LLM API and web environment components for clear identification of bottlenecks. Reveals that both elements significantly impact overall system efficiency across 15 models and 5 providers.

2. Demonstrates high variability in LLM API response times, with differences up to 69.21x for identical requests. Examines factors like model size, service tier, query date, and output token length influencing latency. Highlights persistent variance across dates and geographic locations for consistent insights.

3. Identifies web environment latency as a major contributor, up to 53.7% in benchmarks like WebWalkerQA. Analyzes performance characteristics including fetch latencies and large subpage spaces. Provides empirical foundation for targeted optimizations in agentic systems.

4. Proposes SpecCache as an innovative caching framework augmented with speculative execution. Employs a draft model to predict and prefetch actions asynchronously. Enables overlap of environment interactions with main model reasoning without interfering with the primary path.

**Weaknesses:**

1. The experiment scale seems to be limited and unclear. Firstly, only 30 tasks for web environment analysis and 10 questions per benchmark for evaluations. Secondly, could author clearly illustrate the test setup? Totally 20 questions are used? Or, 30 tasks are used? Could authors explain why this experiment scale can represent the web agent efficiency?

2. This paper evaluates on limited benchmarks like WebWalkerQA and Frames. Thus, the paper title seems to be too general, which is a little over-claimed. The focus of this paper is web agent, or say web search agent.


3. Introduces additional compute overhead from the draft model without detailed quantification. Mentions trade-offs but lacks analysis on cost implications for deployment.

4. This work assumes API-based access to closed-source models, ignoring self-hosted open-source models. Thus, it is unknown whether the findings can be transferred to non-API users.

5. Compares minimally to alternative caching strategies, mentioning naive approaches without experiments. Relies on speculative execution but lacks ablation on components like different cache policy algorithms.

**Questions:**

See weaknesses.

---

### Official Review · Reviewer_FkqQ · 2025-10-25

**Soundness:** 2
**Presentation:** 2
**Contribution:** 3
**Rating:** 4
**Confidence:** 3

**Summary:**

The paper tackles a practical problem: agentic systems are often frustratingly slow. The authors first do a thorough job benchmarking this latency, pinning the blame on two main culprits: the unpredictable lag from LLM APIs and the time spent waiting for web pages to load. To address the web environment bottleneck, they propose SpecCache, a system that uses a secondary, "draft" LLM to predict and pre-fetch the main agent's next actions speculatively. The core idea is to overlap the agent's reasoning time with web interaction time. Their experiments show this approach can reduce the time spent waiting on the web by up to 3.2x, all without altering the agent's final decisions.

**Strengths:**

The analysis across multiple models, providers, dates, and regions (e.g., Figures 2-4, 9-14) offers novel insights into real-world variability, which could inform future system designs.

SpecCache introduces an innovative approach by decoupling reasoning and environment interactions via a draft model for prefetching. This extends speculative decoding concepts to agentic systems, enabling parallelism and potentially generalizing to other turn-based environments (as noted in sec 3.3). The idea of overlapping costs without interfering with the main reasoning path is clever and aligns with efficiency goals.

The inclusion of prompts, trajectories, and sampled questions in appendices enhances reproducibility, and the focus on real-world web interactions (vs. simulated environments) adds ecological validity.

**Weaknesses:**

**Severe Mismatch Between Problem Analysis and Solution:** The paper dedicates substantial space (e.g., sec 2.1 and Appendix E) to analyzing LLM API latency as a major, highly variable bottleneck. However, the proposed SpecCache only addresses web environment latency, while API issues are dismissed by relying on external features like OpenAI's priority processing (not an author contribution). This creates a disjointed structure: the analysis identifies two bottlenecks (API and web), but the solution assumes the former will be resolved by others. The plural "Limits" in the title is misleading, more rigorously, the paper should either address all identified bottlenecks or discuss their combined impact.

**Complete Oversight of Solution Costs:** The paper's focus on "efficiency" is undermined by ignoring the substantial costs of SpecCache. It introduces a draft model (e.g., GPT-4.1-mini alongside o4-mini or GPT-5-mini), effectively doubling LLM API calls per step for parallel execution. Additionally, speculative actions (e.g., 3 per prediction) multiply network I/O requests. Claiming 3.2× latency reductions without a cost-benefit analysis (e.g., monetary or computational overheads increasing by 100-200%) is unacceptable in an efficiency-oriented work . The trade-offs need to be quantified.

**Missing Evaluation of Core Mechanism:** SpecCache's effectiveness hinges on the draft model's prediction accuracy (i.e., cache hit rate). However, the paper reports no data on hit rates, such as the probability that the draft model (GPT-4.1-mini) correctly predicts the target model's next action or the Hit Rate@3 for its three speculative actions. Without this, the 3.2× latency reductions are unexplained and irreproducible, reducing the results to a "black box" (sec 3.2 vaguely mentions accuracy without evidence).

**Insufficient Baseline Comparisons:** Evaluations (sec 4.2) only compare "w/o cache" vs. SpecCache, which is merely an ablation study. Stronger baselines are needed, such as non-LLM prefetching (e.g., heuristic-based link prefetching via keywords) or simpler draft models (e.g., lightweight distilled models like Distill-Qwen-1.5B).

**Questions:**

Why not extend SpecCache to address LLM API latency (e.g., via speculative token generation)?

Sec 3.3 mentions generalization to other agentic systems, can you provide preliminary evidence or examples beyond web environments?

---

### Official Review · Reviewer_58zc · 2025-10-30

**Soundness:** 3
**Presentation:** 3
**Contribution:** 1
**Rating:** 2
**Confidence:** 4

**Summary:**

The paper introduce an engineering design SpecCache to allow environment overhead to be overlapped with LLM reasoning. They also introduce a draft model to propose speculative action so that the environment observation could be prefetched before the target model finish reasoning about next action. SpecCache reduces web environment latency by up to 3.2×.

**Strengths:**

* The paper motivates the techniques with a good analysis on overhead.

* The proposed technique is important to make Web-based system practical.

**Weaknesses:**

* The proposed technique is not so much principled and fundamental motivated but more of an engineering design and product-centric. The paper might be a better fit for HCI conference.

* I have hard time making connection about this work's contribution to future follow up research.

**Questions:**

N/A

---

### Official Review · Reviewer_MS66 · 2025-10-31

**Soundness:** 4
**Presentation:** 3
**Contribution:** 3
**Rating:** 4
**Confidence:** 3

**Summary:**

This paper presents a comprehensive empirical study on the efficiency of web-interactive agentic systems, arguing that research has largely neglected system latency in favor of reasoning performance. The authors decompose the end-to-end latency into two main components: LLM API latency and web environment latency.

Through a robust benchmark of 15 models from 5 providers, they demonstrate that both components are significant bottlenecks. They find high variability in API latency (up to 69.21x variance) and show that web environment interactions can account for up to 53.7% of the total agent iteration time.



To address the web environment bottleneck, the authors propose SpecCache, a caching framework based on speculative execution. SpecCache uses a smaller "draft model" to run in parallel with the main "target model". This draft model predicts the target model's likely future actions (e.g., web clicks) and executes them proactively, storing the results in a cache . If the target model's chosen action is in the cache, it bypasses the web latency entirely . Evaluations on the WebWalkerQA and Frames benchmarks show that SpecCache can reduce web environment overhead by up to 3.2x without compromising the final task performance.

**Strengths:**

The focus on efficiency and latency (not just reasoning accuracy) is highly relevant and addresses a major barrier to the real-world deployment of agentic systems .


The paper's first half (§2) is a robust and useful benchmark of the current landscape of LLM API and web environment latency, providing clear evidence for where the time is spent .


SpecCache is a well-designed solution that directly targets the largest-identified bottleneck (web latency). The idea of overlapping web-fetching with target model reasoning by using a draft model is simple to understand and demonstrably effective.


 A key strength of the proposed architecture is that it accelerates the system without altering the target model's final output or reasoning path, as the speculative actions are non-interfering.

**Weaknesses:**

Your primary focus is on "efficiency" defined as latency, but it completely omits the "efficiency" defined as cost. SpecCache introduces at least one additional LLM (the draft model) running in parallel. The draft model prompt (Table 8) even suggests generating 3 candidate actions, which could triple the draft model's inference cost. This means the system, while faster, could be 2-3x more expensive in terms of API calls and token usage. This cost-latency trade-off is a critical aspect of "efficiency" and its absence is the paper's most significant weakness.

The "up to 3.2x" speedup claim is for the web environment overhead only, not the total end-to-end latency. The end-to-end charts (e.g., Figure 8) show more modest, though still significant, total speedups (e.g., Q1 in Fig 8a goes from ~9s to ~6s, a ~1.5x total speedup). This is a bit of "up-to" marketing and could be presented more transparently.


The success of SpecCache hinges on the draft model's ability to accurately predict the target model's action. The experiments use a very strong draft model (GPT-4.1-mini) for strong target models (o4-mini, GPT-5-mini). The paper does not explore the sensitivity of the cache-hit rate to this pairing. How well would it work if the draft model were a much smaller, less-capable open model (e.g., Llama-3.1-8B) trying to predict a GPT-5-mini? The performance would likely degrade significantly.

**Questions:**

1. Could you please provide data on the computational/cost overhead of SpecCache? Specifically, what is the increase in total tokens processed (or API cost) from running the parallel draft model, and how does this compare to the latency saved? A 2x increase in cost for a 1.5x speedup might be an undesirable trade-off in many applications.

2. The draft model is prompted to produce 3 candidate actions. How often was the target model's actual action in this set (i.e., what was the top-3 accuracy)? Did generating 3 actions add significant latency to the draft model's own inference step?

---

### Meta-Review · Area_Chair_qFPD · 2026-01-06

**Summary:**

This paper presents an empirical study on efficiency bottlenecks in web-interactive agentic systems, decomposing end-to-end latency into LLM API latency and web environment latency. The authors benchmark 15 models across 5 providers, reporting that web latency can account for up to 53.7% of total iteration time. To address this, they propose SpecCache—a caching framework using a draft model to speculatively prefetch web actions in parallel with the main model’s reasoning—claiming up to 3.2× reduction in web environment overhead without affecting task performance.

Reviewers generally agree that the problem is timely and the empirical analysis of latency components is valuable. However, major concerns were raised regarding: (1) the omission of cost/monetary overhead introduced by the draft model; (2) the limited scope of the solution (only addressing web latency despite identifying API latency as a co-primary bottleneck); (3) lack of core metrics such as cache hit rate or prediction accuracy of the draft model; (4) small evaluation scale and narrow benchmark coverage; and (5) insufficient comparison to alternative baselines. One reviewer questioned the paper’s fit for a machine learning venue due to its engineering-oriented contribution.

Notably, the authors did not submit a rebuttal, so none of these concerns were addressed during the discussion phase.

**Reviewer Concerns:**

Since no rebuttal was submitted, none of the substantive technical or methodological concerns were addressed. Key outstanding issues include:

- Absence of cost analysis (e.g., increased API calls, token usage, monetary expense) from running a parallel draft model—highlighted by Reviewers MS66, FkqQ, and bymT.
- Discrepancy between the broad title (“What Limits Agentic Systems Efficiency?”) and the narrow solution that only targets web latency while leaving API latency unaddressed—raised by Reviewer FkqQ.
- Missing evaluation of the draft model’s prediction accuracy or cache hit rate, which is central to interpreting the claimed speedups—emphasized by Reviewer FkqQ.
- Limited evaluation scale (e.g., ~20–30 tasks total) and use of only two benchmarks, raising questions about generalizability—pointed out by Reviewer bymT.
- Lack of comparison to non-LLM-based prefetching or simpler caching baselines—also noted by Reviewer FkqQ.

**Reviewer Scores:**

The ratings are 4,2,4,2. No rebuttal posted.

---

### Decision · Program_Chairs · 2026-01-26

Reject